# The Predictive Power of Barotrauma from the Macklin Effect in the ARDS Population: A Comparison of COVID-19 and Non-COVID-19 ARDS—Could the Macklin Effect Serve as a Helpful Tool for Evaluating Transfer to ARDS Reference Centers?

**DOI:** 10.3390/diagnostics15192514

**Published:** 2025-10-03

**Authors:** Alberto Marabotti, Filippo Pelagatti, Gianluca Frezzetti, Marco Albanesi, Antonio Galluzzo, Alessandra Valletta, Laura Arianna Sorrentino, Andrea Cardoni, Giovanni Cianchi, Marco Ciapetti, Chiara Lazzeri, Adriano Peris, Manuela Bonizzoli

**Affiliations:** 1Intensive Care Unit, Regional ECMO Referral Centre, Azienda Ospedaliero-Universitaria Careggi, 50134 Florence, Italy; pelagattif@aou-careggi.toscana.it (F.P.); alessandra.valletta.av@gmail.com (A.V.); laura.ariannasorrentino@gmail.com (L.A.S.); cardonia@aou-careggi.toscana.it (A.C.); giovannicianchi@gmail.com (G.C.); ciapettim@aou-careggi.toscana.it (M.C.); lazzeric@aou-careggi.toscana.it (C.L.); adriano.peris@gmail.com (A.P.); bonizzolim@aou-careggi.toscana.it (M.B.); 2Department of Emergency Radiology, University Hospital Careggi, Largo Brambilla 3, 50134 Florence, Italy; frezzettig@aou-careggi.toscana.it (G.F.); albanesima@aou-careggi.toscana.it (M.A.); antoniogalluzzo17@gmail.com (A.G.)

**Keywords:** ARDS, Barotrauma, Macklin effect, Invasive mechanical ventilation, COVID-19

## Abstract

**Background:** The Macklin effect recently demonstrated a high positive predictive value for barotrauma in the COVID-19 ARDS population. However, there was less evidence available regarding the ARDS population without SARS-CoV-2 infection. We aim to analyze COVID-19 and non-COVID-19 ARDS subjects to assess the sensitivity and specificity of the Macklin effect in predicting the development of barotrauma in both groups. **Methods:** We retrospectively analyzed subjects with ARDS admitted to our center from January 2018 to November 2022. Experienced radiologists examined the presence of the Macklin effect on chest computed tomography scans. Subjects were then divided into two groups based on the presence or absence of the Macklin effect to assess its predictive power regarding barotrauma. Finally, we analyzed the impact of the Macklin effect and barotrauma on Intensive Care Unit and in-hospital mortality. **Results:** We analyzed 225 patients; the Macklin effect was observed in 44 subjects. In our cohort, the Macklin effect exhibited a sensitivity of 44.6% and a specificity of 90.6% in predicting barotrauma. After excluding the COVID-19 ARDS cases, the Macklin effect showed a sensitivity of 34.7% and a specificity of 93.6%. Nonetheless, in our population, the presence of the Macklin effect or the occurrence of barotrauma did not lead to increased ICU or in-hospital mortality. **Conclusions:** Our analysis highlighted that the Macklin effect demonstrates high specificity in predicting barotrauma but a low sensitivity; moreover, the development of barotrauma did not impact mortality, possibly due to the exclusion of mild to moderate ARDS and the inclusion of a significant number of ECMO recipients. Finally, the Macklin effect appears early during ARDS and may serve as an early indicator of lung frailty, potentially becoming an additional criterion for referral to centers for advanced ARDS treatment.

## 1. Introduction

Acute respiratory distress syndrome (ARDS) is an acute respiratory illness with a high incidence in critical care patients, from 10% of all ICU patients to 23% of all mechanically ventilated patients [1]. ARDS is associated with a high mortality, ranging from 30% to 45% [1,2]. Despite protective strategies, early pronation, and neuromuscular blockade, positive-pressure ventilation could concur to boost lung injury and hamper lung recovery [3]. Barotrauma remains a significant issue in patients with ARDS undergoing positive-pressure mechanical ventilation. Pneumothorax (PNX) and pneumomediastinum (PMD) are relatively frequent in mechanically ventilated patients with ARDS, and the recent COVID-19 pandemic appears to have further increased the incidence of this complication [4]. It is noteworthy that patients with COVID-19 ARDS encountering these complications have a mortality rate of 60% [5]. No specific tests are available to identify individuals at a high risk of barotrauma to justify an ultra-protective ventilation strategy and an early recourse to extracorporeal membrane oxygenation (ECMO) to minimize the risk of ventilator-induced lung injury [6].

The Macklin effect (ME) was first described in 1939 as a sign of pneumomediastinum resulting from alveolar leakage. The released air dissected the lung interstitium along the broncho-vascular sheaths in a centripetal direction towards the pulmonary hilum, reaching the mediastinum [7]. During the COVID-19 pandemic, this radiologic sign regained success. Belletti et al., in their analysis of 124 patients with COVID-19 ARDS, found that 95% of patients with PNX or PMD had the Macklin effect on chest computed tomography [8]. Subsequently, Paternoster and colleagues analyzed 981 patients with COVID-19 pneumonia and identified the Macklin effect in 33 patients (4.7%). Thirty-two of the thirty-three patients with positive ME developed barotrauma. In their analysis, the Macklin effect achieved a positive predictive value (PPV) of 96.7% (95% CI: 80.8–99.5) in predicting barotrauma [9].

We aim to assess the predictive value for barotrauma of the Macklin effect in both COVID-19 and non-COVID-19 ARDS, as well as the influence of the Macklin effect and barotrauma on patient outcomes.

## 2. Methods

This single-center study was conducted at Careggi University Hospital in Florence, an Italian tertiary care center. The Ethical Committee of our center, “Comitato Etico Area Vasta Centro”, approved the study protocol on 26 September 2023, protocol number 23650_oss.

We retrospectively analyzed patients diagnosed with ARDS, according to Berlin’s definition [10], admitted to our ICU between January 2018 and November 2022.

The inclusion criteria were as follows:Age >18 years;The need for invasive ventilation or non-invasive ventilation;The availability of at least a chest CT scan.

The exclusion criteria were as follows:
The genesis of post-procedure pneumothorax or pneumomediastinum (central venous line insertion, tracheostomy, and thoracic drainage);Evidence of pneumothorax or pneumomediastinum at the first available chest CT scan.

Continuous positive airway pressure (CPAP) and pressure support ventilation (PSV) were considered as non-invasive ventilation. Patients received non-invasive ventilation through an oro-nasal or full-face mask; no helmets were used in our intensive care unit. Subjects who required invasive mechanical ventilation received protective ventilation (6–7 mL/kg tidal volume, plateau pressure <27 mmHg, and driving pressure <15 mmHg), neuromuscular blockade, and pronation according to gas exchange and medical judgment. Patients received veno-venous ECMO according to EOLIA trial criteria [11] and medical judgment.

We defined barotrauma as the occurrence of pneumothorax or pneumomediastinum during non-invasive or invasive ventilation, excluding all iatrogenic cases to focus on barotrauma related to lung frailty.

Three experienced radiologists, blinded to the potential development of barotrauma, analyzed the Macklin effect in terms of ‘presence/absence’ and its topographical distribution within the lungs (adjacent to peripheral versus central bronchial branches). The Macklin effect is a linear collection of air at the level of the bronchovascular bundles, visceral pleura, and/or interlobular septa visible on chest CT. Experienced radiologists easily recognize it without the need for special CT scan reconstruction techniques. [12].

The radiologists reviewed all the CT scans performed during the hospitalization for every participant, with particular focus on the first available CT scan to assess the precocity of the Macklin effect appearance.

We investigated differences in the Macklin effect and barotrauma genesis according to age, gender, oxygen gas exchange (Horowitz index), time from symptoms’ onset to ICU admission, and invasive and total (non-invasive and invasive) ventilation length. Finally, we examined the impact of the Macklin effect and barotrauma genesis on ICU and in-hospital mortality.

### Statistical Analysis

The groups’ characteristics were presented as mean and standard deviation (SD). We used the *t*-test to compare numerical variables with a normal distribution and the Mann–Whitney U test for values without a normal distribution. The Kolmogorov–Smirnov test was used to evaluate the normality of the distribution.

We employed Fisher’s exact test to compare categorical variables.

The significance level was set at *p* < 0.05 for each analysis.

Sensitivity, specificity, and positive and negative likelihood ratios were calculated for the Macklin effect in both COVID-19 and non-COVID-19 ARDS groups.

## 3. Results

We examined 239 subjects and ultimately analyzed 225: 9 patients were excluded due to the presence of pneumomediastinum or pneumothorax at the initial CT scan, while 5 others were excluded for post-procedural pneumothorax. The average age of our population was 61 years (SD 13.2), with 159 males (71%). The mean Horowitz index was 104 (SD 41.9), with 91 subjects receiving veno-venous ECMO. All patient characteristics are listed in Table 1.

The Macklin effect was identified in 44 subjects (19.5%), with no differences in sex, etiology (COVID-19 versus non-COVID-19), or the Horowitz index at ICU admission (Table 2). We did not find differences in ICU and in-hospital mortality (45.4% versus 45.3% and 47.7% versus 48%, respectively) between patients with and without the Macklin effect. Twenty-nine subjects with the Macklin effect (66%) developed barotrauma versus thirty-six subjects without radiological Macklin effect signs (20%), *p* < 0.001. The Macklin effect (Figure 1 and Figure 2) demonstrated a sensitivity of 44.6% and a specificity of 90.6% in predicting barotrauma, with a positive likelihood ratio of 4.64 and a negative likelihood ratio of 0.61. We found no differences in the length of NIV before invasive mechanical ventilation (5.7 days [5.1] versus 6.3 days [5.8], *p* = 0.764), length of mechanical ventilation before cannulation in ECMO recipients (6 days [4.3] versus 5 days [3.8], *p* = 0.471), length of NIV in ECMO recipients (12 days [7.9] versus 9 [5.9], *p* = 0.158), or in the length of ventilation before barotrauma genesis (17 days [18.1] versus 17 [21.8], *p* = 0.818) in the Macklin group compared to the non-Macklin effect group.

Excluding COVID-19 ARDS, we analyzed twelve subjects in the Macklin group and seventy-four in the non-Macklin effect group. Eight subjects (66%) developed barotrauma in the Macklin group versus fifteen (20%) in the non-Macklin effect group, *p* < 0.01. In the non-COVID-19 ARDS population, the Macklin effect demonstrated a sensitivity of 34.7% and a specificity of 93.6% in predicting barotrauma, with a positive likelihood ratio of 5.42 and a negative likelihood ratio of 0.70. We found no differences in the length of NIV before invasive mechanical ventilation (3.3 days [3.1] versus 3.5 days [3.6], *p* = 0.849), length of mechanical ventilation before cannulation in ECMO recipients (5 days [4.6] versus 4 days [3.6], *p* = 0.960), or in the length of ventilation before barotrauma genesis (12.5 days [13.1] versus 6.1 [6], *p* = 0.663) in the Macklin group compared to the non-Macklin effect group. However, the group with a positive Mackin effect treated with ECMO received longer NIV before extracorporeal support: 16 days [11.2] versus 7 days [4.6], *p* = 0.012.

The appearance of barotrauma (Figure 3) was identified in 65 subjects (28.8%): 33 subjects with pneumothorax, 18 with pneumomediastinum, and 14 with both. We found no differences in sex, etiology (COVID-19 versus non-COVID-19), or the Horowitz index at ICU admission between subjects who experienced barotrauma and those who did not (Table 3). Nevertheless, in ECMO recipients, we found an increase in the duration of non-invasive ventilation before ECMO (12 days [6.4] versus 9 days [7], *p* = 0.02).

Subjects who experienced barotrauma did not demonstrate a higher ICU and in-hospital mortality (52.3% versus 42.5% and 53.8% versus 45.6%, respectively).

## 4. Discussion

Our analysis demonstrated that the Macklin effect identified on the CT scan predicts the development of future barotrauma in ventilated ARDS subjects (66% of barotrauma in the Macklin group versus 20% in the non-Macklin group, *p* < 0.001). We obtained similar results in the non-COVID-19 ARDS population (66% versus 20% in the non-Macklin group, *p* < 0.01).

This data aligns with recent studies on COVID-19 ARDS subjects [8,9,13,14]. Nonetheless, our study revealed a reduced sensitivity in detecting barotrauma, while maintaining a similar specificity compared to the latest meta-analysis [12] (sensitivity of 44.6% versus 90% and specificity of 90.6% versus 95%). This difference may be attributed to the highly variable timing of the first CT scan performed in our population. Indeed, many patients were transferred from other institutions due to our specific expertise in ARDS management (including veno-venous ECMO) and logistical challenges during the pandemic. Consequently, many subjects in our study developed barotrauma before undergoing a chest CT scan, which precludes the evaluation of the presence of the Macklin effect. This selection bias may explain the significant difference in sensitivity highlighted in our study, though we cannot rule out variability linked to different etiologies and ventilation practices. Nonetheless, we did not find substantial differences in the non-invasive and invasive ventilation length before barotrauma genesis, even in the subgroup of ECMO recipients. In the barotrauma group and the Macklin group of non-COVID-19 ARDS treated with veno-venous ECMO, we found only an increase in the duration of NIV.

Scarce evidence exists regarding the incidence and predictive power of the Macklin effect in ARDS subjects without SARS-CoV-2 infection. Our study innovatively investigates the predictiveness of the Macklin effect in non-COVID-19 ARDS, demonstrating similar results between the two populations. Data on non-COVID-19 ARDS obtained in our study included 87 subjects and must be confirmed in larger populations and multicentric studies.

Our study identified a high incidence of barotrauma (28.8%), which demonstrated no correlation with ICU and in-hospital mortality. This data contradicts previous studies on either the general ARDS population [15,16,17] or COVID-19 [18], where the incidence of barotrauma was lower but correlated with an increase in in-hospital mortality. Several factors could explain this peculiar difference. First, we analyzed a population of only severe ARDS subjects (mean P/F ratio of 104 [SD 41.9]), including a significant group of critically ill patients treated with ECMO. Moreover, our patients developed barotrauma much later compared to the reported data on both COVID-19 and non-COVID-19 ARDS: 17 days of ventilation before barotrauma (SD 20.1) in our population versus 3 days (IQR 0-17) in the multicentric study on SARS-CoV-2 ARDS by Serck and colleagues [18] and 3.4 days (SD 4.2) in the multicentric and international cohort study by Anzueto and colleagues [15]. This lateness in the appearance of barotrauma may be linked to the severity of the disease, with lung frailty becoming the leading actor and reducing the impact of ventilator-induced lung injury and patient self-inflicted lung injury. Finally, the presence of only severe ARDS in our cohort may diminish the striking power of barotrauma on mortality, which is affected by many confounding factors. Furthermore, Schnapp et al. noted that less than 2% of the deaths in their population were linked to pneumothorax [16]. In conclusion, we might infer that barotrauma alone may have had less impact on the outcome of critically ill patients with severe ARDS.

Whether the Macklin effect can predict the progression of early ARDS to a more severe form has not been explored in the literature and is beyond the scope of our study, as we only analyzed severe ARDS patients. Similarly, further research is needed to determine if early ECMO treatment could prevent complications and reduce mortality in patients with the Macklin effect. 

## 5. Conclusions

The Macklin effect observed on CT scans documents a dynamic process occurring due to alveolar leakage, where the released air dissects the lung interstitium along the broncho-vascular sheaths towards the pulmonary hilum in a centripetal direction up to the mediastinum in various forms.

The data from our work reasonably suggest that the Maklin effect develops early in ARDS, both with invasive and non-invasive ventilation. Therefore, performing an early chest CT scan may highlight an ongoing evolutionary process and recommend anticipatory referrals to centers for advanced ARDS treatment. Further investigation is needed to determine whether the Macklin effect may serve as an additional risk factor in the criteria for transfer to ECMO centers.

## Figures and Tables

**Figure 1 diagnostics-15-02514-f001:**
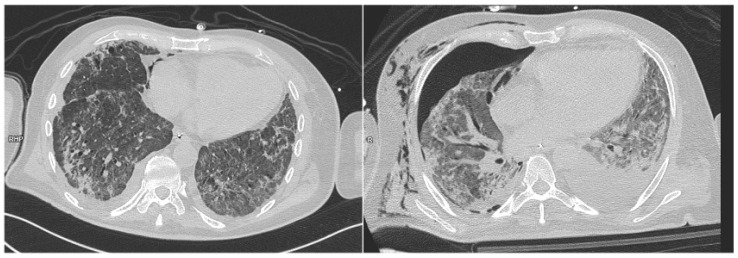
A patient exhibiting the radiological signs of the Macklin effect (**left**) later experienced barotrauma resulting in significant pneumothorax (**right**).

**Figure 2 diagnostics-15-02514-f002:**
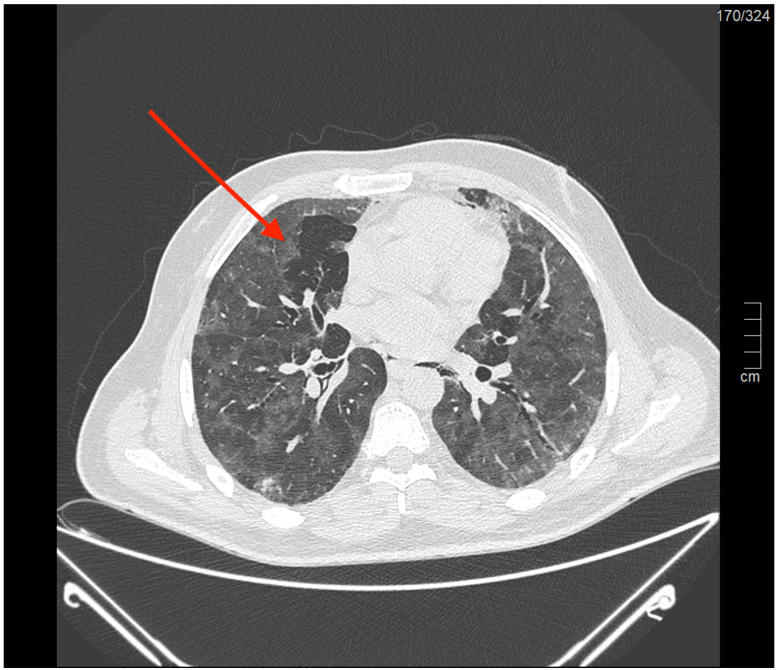
The evidence of a peribronchial Macklin effect (*red arrow*) in an axial lung computed tomography scan.

**Figure 3 diagnostics-15-02514-f003:**
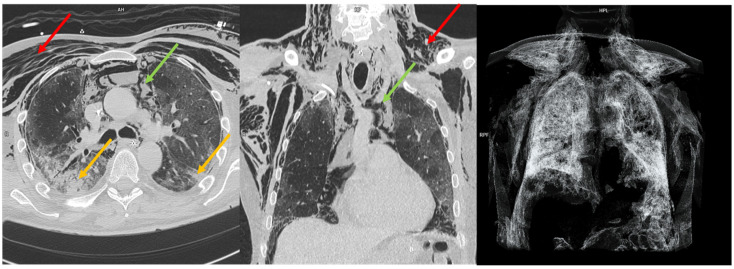
Subcutaneous emphysema extending from the cervical region along the thoracic walls (red arrow) in a COVID-19-positive patient is shown in both axial (**left**) and coronal (**center**) reconstructions. A large pneumomediastinum is present (green arrow). Bilateral confluent areas of parenchymal consolidation (yellow arrow) are also observed. A volume-rendered technique reconstruction is displayed (**right**).

**Table 1 diagnostics-15-02514-t001:** Patients’ characteristics.

Characteristics	N (%) or Mean (±SD)
**Total patients**	225 (100%)
**Males**	159 (71%)
**Females**	66 (29%)
**Age (years)**	61 (±13.3)
**Horowitz index at ICU admission**	104 (±41.9)
**Time from symptoms’ onset to ICU admission (days)**	10 (±9)
**Days of NIV before invasive ventilation**	6 (±5.7)
**Days of NIV before ECMO**	10 (±6.9)
**Days from intubation to ECMO**	5 (±4)
**Days of ventilation before barotrauma**	17 (±20.1)
**Number of subjects with barotrauma**	65 (28.8%)
**ICU mortality**	102 (45.3%)
**In-hospital mortality**	108 (48%)
**Aetiology**	
**COVID-19**	139 (62%)
**Non COVID-19**	86 (38%)
**H1N1**	38 (16%)
**Bacterial**	26 (12%)
**Unknown**	8 (4%)
**Septic shock**	4 (2%)
**HELLP syndrome**	2 (1%)
**ANCA vasculitis**	1 (0.5%)
**Drug overdose**	1 (0.5%)
**Pancreatitis**	1 (0.5%)
**Endocarditis**	1 (0.5%)
**Antisynthetase syndrome**	2 (1%)
**Acute Myeloid Leukemia**	1 (0.5%)
**Dermatomyositis**	1 (0.5%)

Data presented as mean (standard deviation) or number (percentage). ICU: intensive care unit; NIV: non-invasive ventilation; ECMO: extracorporeal membrane oxygenation; HELLP: Hemolysis, Elevated Liver Enzymes and Low Platelets Syndrome; and ANCA: antineutrophil cytoplasmic antibodies.

**Table 2 diagnostics-15-02514-t002:** Data of the Mackin and non-Macklin groups of subjects.

	Macklin Group	Non-Macklin Group	*p*-Value
**Number of subjects**	44	181	
**Males**	34 (77%)	125 (69%)	0.283
**COVID-19**	32 (73%)	107 (59%)	0.095
**Horowitz index at ICU admission**	98 (±21.7)	106 (±45.3)	0.888
**Barotrauma**	29 (66%)	36 (20%)	<0.001
**Length of NIV before invasive mechanical ventilation**	5.7 days (±5.1)	6.3 days (±5.8)	0.764
**Length of mechanical ventilation before cannulation in ECMO recipients**	6 days (±4.3)	5 days (±3.8)	0.471
**Length of NIV before cannulation in ECMO recipients**	12 days (±7.9)	9 (±6.5)	0.158
**Length of ventilation before barotrauma genesis**	17 days (±18.1)	17 (±21.8)	0.818
**ICU mortality**	20 (45.4%)	82 (45.3%)	0.985
**In-hospital mortality**	24 (47.7%)	87 (48%)	0.967

Data presented as mean (standard deviation) or number (percentage). ICU: intensive care unit; NIV: non-invasive ventilation; and ECMO: extracorporeal membrane oxygenation.

**Table 3 diagnostics-15-02514-t003:** Data of the barotrauma and non-barotrauma groups of subjects.

	Barotrauma Group	Non-Barotrauma Group	*p*-Value
**Number of subjects**	65 (29%)	160 (71%)	/
**Males**	48 (74%)	111 (69%)	0.504
**COVID-19**	42 (64.6%)	97 (60%)	0.576
**Horowitz index at ICU admission**	101 (±28.6)	106 (±46.1)	0.920
**Length of NIV before invasive mechanical ventilation**	6.1 (±6)	6.2 (±6)	1.000
**Length of mechanical ventilation before cannulation in ECMO recipients**	6 (±4.5)	5 (±3.6)	0.080
**Length of NIV before cannulation in ECMO recipients**	12 (±6.4)	9 (±7)	0.020
**Length of ventilation before barotrauma genesis**	13 (±15.2)	19 (±22.4)	0.271
**ICU mortality**	34 (52.3%)	68 (42.5%)	0.180
**In-hospital mortality**	35 (53.8%)	73 (45.6%)	0.253

Data presented as mean (standard deviation) or number (percentage). ICU: intensive care unit; NIV: non-invasive ventilation; and ECMO: extracorporeal membrane oxygenation.

## Data Availability

The data presented in this study are available on request from the corresponding author due to ethical restrictions.

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
