# Peer review of "The Predictive Power of Barotrauma from the Macklin Effect in the ARDS Population: A Comparison of COVID-19 and Non-COVID-19 ARDS—Could the Macklin Effect Serve as a Helpful Tool for Evaluating Transfer to ARDS Reference Centers?"

_diagnostics, 2025, doi:10.3390/diagnostics15192514_

Round 1

Reviewer 1 Report

Comments and Suggestions for Authors

Summary

The authors assert that the Macklin effect has demonstrated a high PPV for barotrauma in the COVID-19 ARDS population. Citing limited evidence, their aim was to analyse COVID-19 and non - COVID-19 ARDS subjects in order to assess the sensitivity and specificity of the Macklin effect (ME) in both groups. Their design was a retrospective analysis of ARDS subjects admitted to their centre from January 2018-November 2022. Radiologists looked for the presence of the ME on chest CTs, and divided subjects into two groups based on the presence of absence of ME to assess its predictive value on barotrauma. They also analysed the effect of ME and barotrauma on ICU admission and in-hospital mortality. 

Of 225 patients, they observed ME in 44 subjects (19.6%). They report an ME sensitivity of 44.6% and specificity of 90.6% in predicting barotrauma. When excluding COVID-19 ARDS cases, ME had a sensitivity of 34.7% and specificity of 93.6%. They report that neither ME nor barotrauma lead to increased ICU or in-hospital mortality. 

They conclude that the ME demonstrated high specificity in predicting barotrauma but low sensitivity, and that barotrauma did not affect mortality, possibly due to the exclusion of mild to moderate ARDS and the inclusion of ECMO recipients. Finally, they conclude that ME appears early in ARDS and might be an early indicator of lung frailty, thereby becoming a criterion for early ARDS treatment referral.  

General comments 

In their introduction, the authors provide a brief overview of barotrauma and its association with positive pressure ventilation (PPV), the high association of ARDS amongst COVID-19 patients with such complications and a high mortality rate (60%), and point out that no specific tests exist to a priori identify at-risk patients. They report that the ME (a sign of pneumomediastinum due to alveolar leakage), first described in 1939, regained notoriety during the pandemic, when Belletti et al observed ME on chest CT in 95% of COVID-19 patients with PNX or PMD, and when a subsequent study observed a PPV using ME of 96.7% for barotrauma. Thus, they aimed to assess its PPV in both COVID-19 and non - COVID-19 ARDS, as well as the ME and barotrauma on patient outcomes. 

In the methods, the authors state that this was a single-centre study at a tertiary care centre in Florence, where ethical approval was obtained. Patients were retrospectively diagnosed if they were admitted to the ICU between January 2018 to November 2022, according to their listed inclusion criteria. They provide their definition of barotrauma, and describe their assignment of three blinded experienced radiologists who assigned the presence or absence of the Macklin effect. They describe their statistical analysis procedures for evaluating distribution normality, categorical variables and the other parameters previously described. 

The results section displays the characteristics of the 225 included patients in table 1. Table 2 shows the characteristics of the 44 patients with ME. Figures 1 and 2 display their findings with ME and PNX, with examples of subcutaneous emphysema in a COVID-19 positive patient in figure 3. The results section is laid out very similar to the analysis plan and is easy to follow. One observation by this reviewer is that, although it is true that no significant differences were found in patients with barotrauma vs none in either ICU or in-hospital mortality in each case the percentages are higher, and this may be due to the analytical power of a relatively small study sample size. A larger or even multi-site study with the same findings might have found these differences to be statistically significant, and further underscore the importance of barotrauma on outcomes. This might explain why their findings contradicted other studies’ findings. The authors discuss this in the discussion and provide reasonable explanations for these differences, which should guide any future studies.  

Nonetheless, as the authors point out in the discussion, a CT finding of ME predicts the development of future barotrauma, and validates both the obtaining of early chest CTs, and the sharing of their study findings, as this could contribute to improved clinical management of similar patients.  

Specific comments 

  1. In the abstract page 1, lines 17-18, the authors write, “…and estimate the impact of barotrauma and the Macklin effect on the outcome”. Given that they begin the sentence with their aim to “…assess the sensitivity and specificity of the Macklin effect in both groups”, it is not clear what other outcome they are referring to, apart from positive or negative predictive value. The sentence is complete with “…effect on both groups”, or can be modified to include the predictive values. 
  2. Page 3, lines 103-5; the study aims are provided. This can be omitted as it is previously stated in the introduction. 

Author Response

We thank the reviewer for his careful reading of the proposed article and for his appreciated suggestion. The review process raised several valid questions.

COMMENT 1:

The authors assert that the Macklin effect has demonstrated a high PPV for barotrauma in the COVID-19 ARDS population. Citing limited evidence, their aim was to analyse COVID-19 and non - COVID-19 ARDS subjects in order to assess the sensitivity and specificity of the Macklin effect (ME) in both groups. Their design was a retrospective analysis of ARDS subjects admitted to their centre from January 2018-November 2022. Radiologists looked for the presence of the ME on chest CTs, and divided subjects into two groups based on the presence of absence of ME to assess its predictive value on barotrauma. They also analysed the effect of ME and barotrauma on ICU admission and in-hospital mortality.
Of 225 patients, they observed ME in 44 subjects (19.6%). They report an ME sensitivity of 44.6% and specificity of 90.6% in predicting barotrauma. When excluding COVID-19 ARDS cases, ME had a sensitivity of 34.7% and specificity of 93.6%. They report that neither ME nor barotrauma lead to increased ICU or in-hospital mortality. They conclude that the ME demonstrated high specificity in predicting barotrauma but low sensitivity, and that barotrauma did not affect mortality, possibly due to the exclusion of mild to moderate ARDS and the inclusion of ECMO recipients. Finally, they conclude that ME appears early in ARDS and might be an early indicator of lung frailty, thereby becoming a criterion for early ARDS treatment referral. 

 General comments

In their introduction, the authors provide a brief overview of barotrauma and its association with positive pressure ventilation (PPV), the high association of ARDS amongst COVID-19 patients with such complications and a high mortality rate (60%), and point out that no specific tests exist to a priori identify at-risk patients. They report that the ME (a sign of pneumomediastinum due to alveolar leakage), first described in 1939, regained notoriety during the pandemic, when Belletti et al observed ME on chest CT in 95% of COVID-19 patients with PNX or PMD, and when a subsequent study observed a PPV using ME of 96.7% for barotrauma. Thus, they aimed to assess its PPV in both COVID-19 and non - COVID-19 ARDS, as well as the ME and barotrauma on patient outcomes.

In the methods, the authors state that this was a single-centre study at a tertiary care centre in Florence, where ethical approval was obtained. Patients were retrospectively diagnosed if they were admitted to the ICU between January 2018 to November 2022, according to their listed inclusion criteria. They provide their definition of barotrauma, and describe their assignment of three blinded experienced radiologists who assigned the presence or absence of the Macklin effect. They describe their statistical analysis procedures for evaluating distribution normality, categorical variables and the other parameters previously described.

The results section displays the characteristics of the 225 included patients in table 1. Table 2 shows the characteristics of the 44 patients with ME. Figures 1 and 2 display their findings with ME and PNX, with examples of subcutaneous emphysema in a COVID-19 positive patient in figure 3. The results section is laid out very similar to the analysis plan and is easy to follow. One observation by this reviewer is that, although it is true that no significant differences were found in patients with barotrauma vs none in either ICU or in-hospital mortality in each case the percentages are higher, and this may be due to the analytical power of a relatively small study sample size. A larger or even multi-site study with the same findings might have found these differences to be statistically significant, and further underscore the importance of barotrauma on outcomes. This might explain why their findings contradicted other studies’ findings. The authors discuss this in the discussion and provide reasonable explanations for these differences, which should guide any future studies. 

Nonetheless, as the authors point out in the discussion, a CT finding of ME predicts the development of future barotrauma, and validates both the obtaining of early chest CTs, and the sharing of their study findings, as this could contribute to improved clinical management of similar patients. 

 Specific comments

 In the abstract page 1, lines 17-18, the authors write, “…and estimate the impact of barotrauma and the Macklin effect on the outcome”. Given that they begin the sentence with their aim to “…assess the sensitivity and specificity of the Macklin effect in both groups”, it is not clear what other outcome they are referring to, apart from positive or negative predictive value. The sentence is complete with “…effect on both groups”, or can be modified to include the predictive values.

ANSWER 1:

Thank you for your effort in reviewing and for your suggestions. We've changed the sentence to (15-17):

“We aim to analyze COVID-19 and non-COVID-19 ARDS subjects to assess the sensitivity and specificity of the Macklin effect in predicting the development of barotrauma in both groups.”

COMMENT 2:

Page 3, lines 103-5; the study aims are provided. This can be omitted as it is previously stated in the introduction.

ANSWER 2: 

Thank you for your hard work in reviewing and for your suggested corrections. We have removed the sentence.

Reviewer 2 Report

Comments and Suggestions for Authors

This paper demonstrates with actual data that the Macklin effect may aid in detecting ARDS caused by COVID-19 or other causes. As the authors themselves state, it is clear that accumulating data across a wider range of situations is necessary; for that purpose, I believe this data is valuable.

Several issues exist. I would like to see these discussed further, and ideally, this research continued. I consider this a kind of interim report, but one that is essential.

One issue is sensitivity. This paper only utilised patients whose condition had deteriorated sufficiently with quite low Horowitz indexes, and measurements were not taken early enough. This is likely one reason for the low sensitivity. To predict ARDS in advance, this problem will need to be addressed. Is this due to issues with the method used to detect the Macklin effect? Or could AI potentially improve this? I would like to see this point discussed.

Another, more fundamental issue is that even if the Macklin effect is detected, it does not improve mortality rates. Assuming the Macklin effect is present, could ARDS be avoided, for example, by initiating ECMO earlier? If this cannot be achieved, and if the course of action remains unchanged even when ARDS is known to occur, the significance of early detection becomes negligible. This point also requires clear discussion: how might ARDS be prevented?

Additionally, please consider the following points.

Fig. 1: Could you better explain the radiological signs of the Macklin effect? It seems likely that these are difficult to judge without viewing serial sections; please include this in your explanation to readers.

Table 1-3: Please add the percentage symbol (%) to the percentages. They are difficult to read.

Author Response

We thank the reviewer for his careful reading of the proposed article and for his appreciated suggestion. The review process raised several valid questions.

COMMENT 1:

This paper demonstrates with actual data that the Macklin effect may aid in detecting ARDS caused by COVID-19 or other causes. As the authors themselves state, it is clear that accumulating data across a wider range of situations is necessary; for that purpose, I believe this data is valuable.
Several issues exist. I would like to see these discussed further, and ideally, this research continued. I consider this a kind of interim report, but one that is essential.

ANSWER 1: 

Thank you for your valuable contribution and interest in the topic. We fully support ongoing work in this area and see this research as a foundation for future studies. We emphasized this idea in lines 205-209. 

COMMENT 2: 

One issue is sensitivity. This paper only utilised patients whose condition had deteriorated sufficiently with quite low Horowitz indexes, and measurements were not taken early enough. This is likely one reason for the low sensitivity. To predict ARDS in advance, this problem will need to be addressed. Is this due to issues with the method used to detect the Macklin effect? Or could AI potentially improve this? I would like to see this point discussed.
Another, more fundamental issue is that even if the Macklin effect is detected, it does not improve mortality rates. Assuming the Macklin effect is present, could ARDS be avoided, for example, by initiating ECMO earlier? If this cannot be achieved, and if the course of action remains unchanged even when ARDS is known to occur, the significance of early detection becomes negligible. This point also requires clear discussion: how might ARDS be prevented?

ANSWER 2: 

The reviewer has raised important questions for this study.
It is possible that screening the population not yet in ARDS would help prevent its onset in selected cases, but we have not tested the effectiveness of the Macklin effect in this regard. The purpose of this preliminary study was to determine if we could identify a 'super fragile' ARDS phenotype by means of a retrospective analysis. Similarly, our research data do not allow for a proper investigation into whether early ECMO influences the outcome of patients with mild ARDS and a positive Macklin effect.
In this regard, we added the following sentences at the end of the discussion (205-210).
“Whether the Macklin effect can predict the progression of early ARDS to a more severe form has not been explored in the literature and is beyond the scope of our study, as we only analyzed severe ARDS patients. Similarly, further research is needed to determine if early ECMO treatment could prevent complications and reduce mortality in patients with the Macklin effect.”

COMMENT 3: 

Additionally, please consider the following points.
Fig. 1: Could you better explain the radiological signs of the Macklin effect? It seems likely that these are difficult to judge without viewing serial sections; please include this in your explanation to readers.

ANSWER 3: 

Thanks for clarifying this point. Actually, serial sections are not necessary to recognize the Macklin effect. However, a radiologist with expertise in chest CT is required.
In this regard, we changed these sentences (96-99):
From “The Macklin effect is defined as a radiological sign identifiable on chest computed tomography (CT) scans consisting of air dissecting along perivascular and peribronchial interstitial sheaths” to “The Macklin effect is a linear collection of air at the level of the bronchovascular bundles, visceral pleura, and/or interlobular septa, visible on chest CT. Experienced radiologists easily recognize it without the need for special CT scan reconstruction techniques.”

COMMENT 4: 

Table 1-3: Please add the percentage symbol (%) to the percentages. They are difficult to read.

ANSWER 4: 

Tables fixed, thank you for the suggestion.